# Isolation and Quantification of L-Tryptophan from *Protaetia brevitarsis seulensis* Larvae as a Marker for the Quality Control of an Edible Insect Extract

**DOI:** 10.3390/insects16090905

**Published:** 2025-08-29

**Authors:** Hye Jin Yang, Wei Li

**Affiliations:** Korean Medicine (KM) Application Center, Korea Institute of Oriental Medicine, Daegu 41062, Republic of Korea; hjyang@kiom.re.kr

**Keywords:** *Protaetia brevitarsis seulensis* larvae, L-tryptophan, HPLC-DAD, marker compound, quality control

## Abstract

*Protaetia brevitarsis seulensis* (Kolbe, 1886; Coleoptera, Scarabaeoidea)*,* commonly called white-spotted flower chafer larvae and traditionally used in East Asian medicine, are gaining attention as edible insects with functional properties. However, their broader application is limited by the insufficient identification of marker compounds. In this study, a major component of the 70% ethanol extract of *P. brevitarsis* was isolated and identified as L-tryptophan, an essential amino acid. A high-performance liquid chromatography with diode array detection (HPLC-DAD) method was developed for its accurate quantification, confirming that L-tryptophan is a consistently detectable and analytically reliable constituent. These results indicate that L-tryptophan may serve as a quality marker, contributing to the standardization and quality control of insect-derived functional materials.

## 1. Introduction

Edible insects are increasingly recognized as a sustainable source of protein, offering high nutritional value with a lower environmental footprint. Compared to conventional livestock, they require substantially less land, water, and feed and emit fewer greenhouse gases [1,2]. Their protein content may reach up to 77% (composition can vary depending on species, developmental stage, and nutrition) on a dry weight basis and is accompanied by a balanced profile of essential amino acids, vitamins, and minerals [3,4,5]. These characteristics support their potential as an alternative to conventional animal proteins in addressing global nutritional demands. Beyond their macronutrient value, edible insects also contain functional compounds such as peptides, antioxidants, and chitin, which have been linked to physiological benefits, including immune modulation and gut health enhancement [6,7]. It is important to note that chitin may cause allergic reactions. Additionally, insect-derived peptides have been reported to exert antihypertensive, antioxidant, and antimicrobial activities, supporting their potential application as functional food ingredients [8,9].

*Protaetia brevitarsis seulensis* (Kolbe, 1886) larvae are regarded as a representative edible insect and have been traditionally used in East Asian countries, such as Korea and China, for managing conditions including liver disease, urinary dysfunction, breast cancer, and inflammation [10,11,12].

Pharmacological studies have reported that ethanol extracts of *P. brevitarsis* prevent bone loss and inhibit osteoclastogenesis, supporting their anti-osteoporotic potential [13]. Additional findings include neuroprotective effects against trimethyltin-induced damage [14], amelioration of benign prostatic hyperplasia [15], and improvement in nonalcoholic fatty liver disease [16].

Despite increasing interest in its pharmacological effects and industrial potential, chemical studies on *P. brevitarsis* are still limited. Recent research has identified bioactive constituents, including unsaturated fatty acids, amino acid derivatives, and small molecules such as quinoxaline and dopamine derivatives, some of which show tyrosinase inhibitory activity [17,18]. In addition, several indole alkaloids isolated from *P. brevitarsis* larvae have been shown to inhibit platelet aggregation and exert antithrombotic effects, suggesting further therapeutic potential [19]. However, research on the isolation and quantification of specific marker compounds remains insufficient. To date, no constituent has been widely recognized for use in quality control, and standardized protocols for compositional analysis have yet to be established. This may limit the reproducibility and consistency of extract evaluation. Further research is necessary to identify reliable candidates suitable for quantitative assessment and standardization.

In this study, we isolated a primary constituent from *P. brevitarsis*, elucidated its structure, and developed a validated high-performance liquid chromatography with diode array detection (HPLC-DAD) method for its quantification. Given its relatively high yield, well-characterized structure, and precise quantification, this compound is a promising candidate as a quality marker (Q-marker) for the quality control and standardization of *P. brevitarsis*, which is a compound selected to represent the chemical characteristics of a complex biological or natural product for the purpose of quality control and standardization. Q-markers are typically chosen based on their abundance, stability, reproducibility across production batches, and relevance to the biological or functional properties of the material. In this regard, the use of standardized samples produced under Good Manufacturing Practice (GMP) conditions, with demonstrated high reproducibility across independent production batches, is critical to ensuring batch-to-batch consistency and the reliability of quality control procedures. This approach reinforces the validity of compositional analyses and facilitates the establishment of robust quality standards for *P. brevitarsis*. Accordingly, the present work not only addresses the current lack of well-defined marker compounds for this species but also provides a reproducible analytical framework that can support future pharmacological and industrial applications.

## 2. Materials and Methods

### 2.1. General Experimental Procedures

For column chromatography (CC), normal-phase silica gel (Kieselgel 60, 70–230, and 230–400 mesh; Merck, Darmstadt, Germany) was used. Thin-layer chromatography (TLC) was performed on pre-coated silica gel 60 F_254_ and RP-18 F_254_S plates (0.25 mm; Merck, Darmstadt, Germany). A 10% (*v*/*v*) aqueous sulfuric acid (H_2_SO_4_) solution was used as a spray reagent to visualize the TLC spots.

Medium-pressure liquid chromatography (MPLC) was carried out on an Isolera One system fitted with SNAP KP-SIL and SNAP Ultra C18 cartridges (Biotage, Uppsala, Sweden). Recycling preparative high-performance liquid chromatography (Recycling preparative HPLC) was performed using an LC-9130G NEXT instrument equipped with a JAI GEL-GS310 column (20 × 500 mm; Japan Analytical Industry Co., Tokyo, Japan). Proton and carbon nuclear magnetic resonance (NMR) spectra were obtained using a Bruker Avance III 400 MHz spectrometer (^1^H at 400 MHz, ^13^C at 100 MHz; Bruker Biospin GmbH, Rheinstetten, Germany). High-resolution mass spectrometry (HR-MS) was conducted using a Q-Exactive Orbitrap mass spectrometer coupled to an ultra-high-performance liquid chromatography (UHPLC) system and a heated electrospray ionization (HESI) source (Thermo Fisher Scientific, San Jose, CA, USA).

EP-grade solvents used for extraction, fractionation, and isolation were obtained from DAEJUNG Chemicals & Metals Co., Ltd. (Siheung, Republic of Korea). HPLC- and MS-grade methanol (MeOH), acetonitrile (ACN), water, and formic acid (FA) were purchased from Thermo Fisher Scientific (Pittsburgh, PA, USA). Deuterium oxide (D_2_O), the deuterated solvent for NMR analysis, was obtained from Merck (Darmstadt, Germany). The L-tryptophan reference standard (≥98% purity) was obtained from Sigma-Aldrich (St. Louis, MO, USA).

### 2.2. Edible Insect Material

The freeze-dried powder of *P. brevitarsis* larvae used in this study was provided by the National Institute of Agricultural Sciences at the Rural Development Administration (Wanju, Republic of Korea). The production and processing methods for the powder have been described in detail in a previous study [13]. For preparation of the 70% ethanol (EtOH) extract, the freeze-dried powder of *P. brevitarsis* larvae was entrusted to the Food Research Center at the Jeonnam Bioindustry Foundation (Naju, Republic of Korea). Extracts were produced under Good Manufacturing Practice (GMP) conditions (1:10 *v*/*v*; 75 °C; 6 h; one-time extraction; freeze-drying). Reproducibility under the standardized production process was confirmed through analysis of three independent production batches (Appendix A). A voucher specimen (KIOM-AP-LH-407) was deposited in the herbarium of the KM Application Center at the Korea Institute of Oriental Medicine (Daegu, Republic of Korea).

### 2.3. Isolation of the Major Compound

Preliminary HPLC screening of the *P. brevitarsis* ethanol extract (PBE) revealed a prominent peak corresponding to a major compound in the extract. The fractionation and isolation workflow is illustrated in Figure 1. To isolate this major compound, 100 g of the 70% ethanol extract (PBE) was first suspended in distilled water and then successively partitioned with equal volumes of *n*-hexane, ethyl acetate (EtOAc), and *n*-butanol (*n*-BuOH). This procedure yielded the following fractions: *n*-hexane fraction (PBE-H, 1.37 g), EtOAc fraction (PBE-E, 1.40 g), *n*-BuOH fraction (PBE-B, 6.81 g), and the remaining aqueous fraction (58.11 g, PBE-W). Among them, PBE-B, which showed the most prominent peak in HPLC chromatograms, was selected for further isolation. Fraction PBE-B (6.81 g) was further subjected to vacuum liquid chromatography (VLC) on a silica gel column (12.0 × 12.0 cm). A total of six fractions (A–F) were obtained using a stepwise elution with chloroform/methanol (CHCl_3_:MeOH, *v*/*v*) mixtures at 10:1, 8:1, 6:1, 4:1, 2:1, and finally 100% MeOH. Fraction E was further separated by MPLC on a SNAP Ultra C18 cartridge using a gradient elution of MeOH–H_2_O (10–50% MeOH, *v*/*v*), yielding three sub-fractions (Frs. EA–EC). Sub-fraction EB was further purified using recycling preparative HPLC under isocratic elution with MeOH–H_2_O (10% MeOH, *v*/*v*), resulting in the isolation of the major compound (70.0 mg).

The structural elucidation of the isolated major compound was performed using ^1^H and ^13^C NMR spectroscopy. NMR spectra were recorded with a Bruker Avance 500 MHz spectrometer at room temperature, using deuterated methanol (CD_3_OD) as the solvent. Chemical shifts (δ) are reported in parts per million (ppm) relative to the solvent signal, and coupling constants (*J*) are given in Hertz (Hz). The NMR data were used to confirm the chemical structure of the isolated compound.

**L-Tryptophan:** White amorphous powder; C_11_H_12_N_2_O_2_; HR-MS *m*/*z* 205.0973 [M + H]^+^ (calcd. 205.0971); ^1^H NMR (400 MHz, D_2_O) *δ* 7.68 (1H, dt, *J* = 8.1, 1.1 Hz, H-4), 7.49 (1H, dt, *J* = 8.1, 1.1 Hz, H-7), 7.26 (1H, s, H-2), 7.23 (1H, ddd, *J* = 8.1, 7.0, 1.1 Hz, H-6), 7.15 (1H, ddd, *J* = 8.1, 7.0, 1.1 Hz, H-5), 3.99 (1H, m, H-9), 3.44 (1H, dd, *J* = 15.3, 4.8 Hz, H-8), 3.25 (1H, dd, *J* = 15.3, 8.1 Hz, H-8); ^13^C NMR (100 MHz, D_2_O) *δ* 174.5 (C-10), 136.3 (C-7a), 126.6 (C-3a), 125.0 (C-2), 122.1 (C-6) 119.5 (C-5), 118.4 (C-4), 111.9 (C-7), 107.4 (C-3), 55.0 (C-9), 26.4 (C-8).

### 2.4. Preparation of Standard Solutions and Samples

A stock solution of the standard compound was prepared by dissolving it in MeOH to a concentration of 1000 μg/mL. For sample preparation, PBE and PBE-B were dissolved in MeOH at final concentrations of 20 mg/mL and 10 mg/mL, respectively. Prior to analysis, all solutions were filtered through a 0.22 μm polytetrafluoroethylene (PTFE) syringe filter (Sartorius AG, Göttingen, Germany).

### 2.5. Chromatographic Conditions

Quantitative analysis of L-tryptophan in the PBE and PBE-B samples was performed using a Thermo Dionex UltiMate 3000 HPLC system (Thermo Fisher Scientific, San Jose, CA, USA) equipped with a binary pump, auto-sampler, column oven, and diode array detector (DAD). Chromatographic data were collected and processed using the Chromeleon™ 7 chromatography data system (version 7.2.2.6394; Thermo Fisher Scientific, San Jose, CA, USA). Chromatographic separation was carried out on a Thermo Acclaim™ Polar Advantage II column (250 × 4.6 mm, 5 μm; Thermo Fisher Scientific, Sunnyvale, CA, USA). The column temperature was maintained at 30 °C, and the injection volume was set to 5 µL. UV detection was performed at 280 nm. The mobile phase consisted of 0.1% (*v*/*v*) FA in water (solvent A) and MeOH (solvent B), with gradient elution programmed as follows: 0–10 min, 0.2% B; 10–20 min, 0.2–6.5% B; 20–25 min, 6.5–30% B; 25–35 min, 30–100% B; and 35–40 min, 100% B. The flow rate was maintained at 1.0 mL/min. Following each run, the column was re-equilibrated to the initial conditions for 10 min prior to the next injection.

The chromatographic method was validated in accordance with ICH (Q2) guidelines. Standard solutions of L-tryptophan were subsequently diluted in MeOH to obtain final concentrations of 25, 50, 100, 200, and 400 μg/mL. Calibration curves for the quantification of L-tryptophan were prepared using these standards. Linearity over this range was evaluated by calculating the correlation coefficient (r^2^) for each curve. The calibration curves were obtained by plotting the peak area versus the corresponding analyte concentration, and the regression equations were expressed as *y* = *ax* ± *b*, where *y* is the peak area and *x* is the analyte concentration. Quantification of L-tryptophan in the PBE and PBE-B samples was performed based on the external standard method, using the established calibration curve. Each sample was analyzed in quintuplicate (n = 5), and the results were reported as mean ± standard deviation (SD). The limit of detection (LOD) and limit of quantification (LOQ) were calculated based on a signal-to-noise ratio of 3 and 10, respectively.

## 3. Results and Discussion

### 3.1. Analytical Characterization of a Major Compound from PBE

Freeze-dried *P. brevitarsis* larvae powder was extracted with 70% EtOH to obtain a crude ethanol extract (PBE), which was subsequently fractionated using *n*-hexane, EtOAc, and *n*-BuOH, yielding the PBE-H, PBE-E, and PBE-B fractions, respectively. To identify the primary constituents of *P. brevitarsis* larvae, HPLC analysis was performed on the PBE and its fractions. According to the chromatographic profiles presented in Appendix A, both the PBE and PBE-B showed a major peak at approximately 26 min, with PBE-B exhibiting the highest intensity of this peak. These results suggest that the main component of *P. brevitarsis* larvae is predominantly enriched in fraction PBE-B. To further confirm the identity of the major peak, HR-MS analysis was performed to determine its exact molecular mass. The target peak detected in PBE-B yielded a protonated molecular ion at *m*/*z* 205.0973 [M + H]^+^ (calcd. 205.0971 for C_11_H_13_N_2_O_2_), corresponding to the molecular formula C_11_H_12_N_2_O_2_ (Appendix A). The observed *m*/*z* value and the associated elemental composition suggested that the compound was likely an amino acid. The major component of PBE-B, as identified by HPLC and MS analyses, was subsequently isolated and purified.

### 3.2. Isolation and Structural Characterization

Chromatographic separation of PBE-B (6.81 g) by VLC on silica gel yielded six fractions (A–F), among which fraction E was further separated by MPLC to yield three sub-fractions (EA–EC). Subsequent purification of sub-fraction EB by recycling preparative HPLC yielded a single compound (70.0 mg), corresponding to the major peak previously observed in the PBE-B fraction.

NMR spectroscopy was used to determine the chemical structure of the isolated compound. The 1H NMR spectrum (400 MHz, D_2_O) exhibited characteristic signals for an indole moiety, including a singlet at δH 7.26 (H-2), along with four aromatic protons between δH 7.15 and 7.68 (H-4 to H-7), consistent with a five-substituted indole ring. Aliphatic proton signals appeared at δH 3.99 (H-9, methine) and δH 3.44/3.25 (H-8, methylene), corresponding to the α- and β-protons of tryptophan. The 13C NMR data further confirmed the presence of 11 carbon signals, including a carboxylic acid carbon at δC 174.5 (C-10), indole carbons between δC 107.4 and 136.3, and aliphatic carbons at δC 55.0 (C-9) and 26.4 (C-8) (Appendix A). Based on detailed spectroscopic analysis and comparison with reported literature data [17], the compound was identified as L-tryptophan, and its chemical structure is depicted in Figure 1.

### 3.3. Optimization of HPLC Conditions for Quantitative Analysis

Prior to establishing the optimized analytical conditions, the identity of the isolated compound was confirmed by comparing its retention time and UV spectrum with those of the L-tryptophan reference standard using HPLC-DAD analysis. These results are shown in Figure 2, which presents the HPLC chromatograms of the reference L-tryptophan standard (A), the isolated compound (B), the PBE (C), and PBE-B (D). After confirming the identity of the main compound, the same HPLC platform was optimized for its quantitative analysis in the PBE. This procedure was designed to enable the reliable detection of the marker compound and to support the establishment of content criteria for the quality control and standardization of PBE.

Optimized HPLC-DAD conditions for the quantification of L-tryptophan in PBE were established as follows: To improve chromatographic resolution, 0.1% (*v*/*v*) FA was added to the aqueous phase, and MeOH was used as the organic phase under a gradient elution program [20]. FA acidified the mobile phase, enhancing the ionization of L-tryptophan and reducing peak tailing by minimizing interactions between the analyte and the stationary phase. As a result, peak symmetry and resolution were improved, leading to increased detection sensitivity and reproducibility. To further improve the resolution of the highly polar compound L-tryptophan, a Thermo Scientific™ Acclaim™ Polar Advantage II (PA2) column was employed. This column features a polar-embedded stationary phase that enhances selectivity and retention for polar analytes under reversed-phase conditions. It is compatible with 100% aqueous mobile phases and remains stable over a broad pH range (1.5–10), allowing for efficient separation of acidic, basic, and neutral compounds with minimal peak tailing. In this study, the combination of an optimized mobile phase and a PA2 column effectively separated L-tryptophan and improved peak shape and resolution, thereby enhancing analytical sensitivity and reproducibility.

### 3.4. Quantitative Determination of L-Tryptophan in PBE

The developed method enabled accurate quantification and reliable identification of the marker compound in PBE. As shown in Figure 2, it was well separated with excellent resolution within 40 min, and interference from other components was minimal. The analysis was performed at 280 nm, the characteristic absorption wavelength of L-tryptophan. The peak was identified by comparing its retention time (*t*_R_; 26.28 min for standard, 26.27 min for isolated compound, 26.30 min for PBE, and 26.31 min for PBE-B), UV spectrum, and chromatographic profile with those of the standard compound.

As shown in Table 1, L-tryptophan was successfully quantified in the PBE sample using the established calibration curve, which exhibited excellent linearity over the tested range (r^2^ > 0.9999, n = 5). The LOQ and LOD were calculated based on a signal-to-noise ratio (S/N) of 10 and 3, respectively. As shown in Table 1, the calculated LOD was 3.28 μg/mL, and the LOQ was 9.95 μg/mL. The relatively high LOD and LOQ values observed in this study are attributable to the use of UV detection, which is inherently less sensitive than mass spectrometry- or fluorescence detection-based methods [21]. These observations are similar to those of previous studies on amino acids, including L-tryptophan [22]. Nonetheless, UV detection offers advantages in terms of accessibility, cost-effectiveness, and operational simplicity, making it suitable for routine quality control. The RSD at the LOQ level was 1.42%, remaining well within the accepted threshold of 2%, thereby indicating the adequate precision and analytical robustness of the method. These results support the application of the method to quantitative analysis of L-tryptophan in PBE. Based on the resulting regression equation, the content of L-tryptophan was determined to be 1.93 ± 0.05 μg/mg (with the yield of L-tryptophan as 0.193 ± 0.005% of crude extract), with an RSD of 0.72% (n = 3).

The analyte was clearly detected without interference, and the results demonstrated
high reproducibility across replicates.

## 4. Conclusions

In the present study, L-tryptophan was selected as a quality control (QC) marker for *Protaetia brevitarsis seulensis* larval extracts produced under Good Manufacturing Practice (GMP) conditions. This selection was based on repeated fractionation and compositional analyses, which consistently revealed L-tryptophan as the most abundant low-molecular-weight compound with minimal batch-to-batch variation across three independent production lots from a GMP-certified facility designated by the Rural Development Administration (RDA), Republic of Korea. While L-tryptophan is a common amino acid and not unique to *P. brevitarsis*, its high abundance, stability under standardized production conditions, and reproducibility in quantitative analysis make it a practical marker for routine QC testing. In other functional food materials, widely distributed compounds such as resveratrol and chlorogenic acid are similarly used as QC markers when they demonstrate such characteristics. We acknowledge that additional studies are needed to assess its correlation with functional properties, stability under suboptimal processing or storage conditions, and variability across different rearing environments. These aspects will be addressed in future collaborative research.

## Data Availability

The original contributions presented in the study are included in the article/Appendix A. Further inquiries can be directed to the corresponding author.

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
