# Peer review of "Isolation and Quantification of L-Tryptophan from Protaetia brevitarsis seulensis Larvae as a Marker for the Quality Control of an Edible Insect Extract"

_insects, 2025, doi:10.3390/insects16090905_

Round 1
Reviewer 1 Report
Comments and Suggestions for Authors
General Comment
While the study successfully isolates and quantifies L-tryptophan from Protaetia brevitarsis seulensis larvae extracts using HPLC-DAD, the manuscript does not convincingly establish why L-tryptophan should be considered a reliable marker for quality control. The current data show that L-tryptophan is a major compound in the ethanol extract and that it can be quantified accurately, but several key aspects needed to justify its role as a QC marker are missing:
- Biological or processing relevance – There is no evidence that L-tryptophan content correlates with the functional, nutritional, or safety aspects of the extract.
- Stability across production conditions – The study examines a single source and does not test larvae from different diets, developmental stages, or rearing environments to demonstrate that L-tryptophan content remains stable in high-quality extracts.
- Sensitivity to degradation – No experiments are presented to show whether L-tryptophan content decreases under poor processing or storage conditions, which would make it useful as a QC indicator.
- Specificity – L-tryptophan is a common amino acid in many protein sources and is not unique to P. brevitarsis. Without data proving its species-specific profile or consistent enrichment in properly processed extracts, its selection as a unique marker is questionable.
As it stands, the manuscript presents an analytical method for L-tryptophan quantification but does not provide sufficient experimental or contextual evidence to support its use as a robust quality control marker for edible insect extracts. Additional validation across multiple batches, rearing conditions, and potential degradation scenarios would be required to substantiate this claim.
here are small other points
Title and Abstract
-
HPLC is not defined in the abstract.
-
In lines 10 and 21, the full scientific name should be given with the authority and year of description at the first occurrence, along with the family and order, since this insect is not widely known.
Introduction
-
Line 46: Please clarify that composition can vary depending on species, developmental stage, and nutrition.
-
Line 50: Mention that chitin can cause allergic reactions, as this is relevant for edible insect products.
-
Line 55: As above, include the full scientific name with descriptor and year.
-
Lines 65–67: The sentence is heavy and the English should be revised for clarity.
-
Line 77: HPLC-DAD is not defined at first mention.
-
Lines 79–82: This section reads more like results than introduction.
Materials and Methods
-
Line 112: Explain why the extraction starts from ethanol rather than directly from the powdered sample.
-
Line 117: The text implies that this compound is the major one, but there could be 2–3 compounds present in similar proportions. Were preliminary tests performed to justify this assertion? If so, they should be described.
Discussion
-
The discussion does not address the relevance of L-tryptophan as a marker for quality control. This should be elaborated, with reference to practical or regulatory contexts.
-
The study did not test samples from different production conditions. Would the tryptophan profile remain stable under varying rearing or processing conditions? This limitation should be acknowledged.
Author Response
Reviewers' comments to Author:
General Comment
While the study successfully isolates and quantifies L-tryptophan from Protaetia brevitarsis seulensis larvae extracts using HPLC-DAD, the manuscript does not convincingly establish why L-tryptophan should be considered a reliable marker for quality control. The current data show that L-tryptophan is a major compound in the ethanol extract and that it can be quantified accurately, but several key aspects needed to justify its role as a QC marker are missing:
Biological or processing relevance – There is no evidence that L-tryptophan content correlates with the functional, nutritional, or safety aspects of the extract.
Stability across production conditions – The study examines a single source and does not test larvae from different diets, developmental stages, or rearing environments to demonstrate that L-tryptophan content remains stable in high-quality extracts.
Sensitivity to degradation – No experiments are presented to show whether L-tryptophan content decreases under poor processing or storage conditions, which would make it useful as a QC indicator.
Specificity – L-tryptophan is a common amino acid in many protein sources and is not unique to P. brevitarsis. Without data proving its species-specific profile or consistent enrichment in properly processed extracts, its selection as a unique marker is questionable.
As it stands, the manuscript presents an analytical method for L-tryptophan quantification but does not provide sufficient experimental or contextual evidence to support its use as a robust quality control marker for edible insect extracts. Additional validation across multiple batches, rearing conditions, and potential degradation scenarios would be required to substantiate this claim.
In conclusion, while we acknowledge the limitations highlighted by the reviewer—particularly the lack of multi-source validation and degradation studies—the selection of L-tryptophan in this work was based on its high abundance, exceptional batch-to-batch consistency under GMP production, and reproducibility in quantitative analysis. We have revised the manuscript to clarify these points and to explicitly outline our plans for future studies that will address biological relevance, multi-source variability, and degradation sensitivity.
Response to General Comment:
We appreciate the reviewer’s insightful comments regarding the justification of L-tryptophan as a quality control (QC) marker in Protaetia brevitarsis seulensis larval extracts. Below, we address each point in detail.
- Biological or processing relevance
We agree that functional or biological relevance is an important consideration for selecting a QC marker. In our previous studies, P. brevitarsis larval extracts have been evaluated for various bioactivities, including anti-inflammatory and osteoclastogenesis inhibitory effects. While these functional evaluations were performed at the extract level, the current study’s primary objective was to establish, isolate, and quantify a representative marker compound for extracts produced under Good Manufacturing Practice (GMP) conditions, rather than to assess the biological role of the marker itself. We fully acknowledge that correlation studies between L-tryptophan content and functional properties will be valuable and are planning to conduct such investigations in future collaborative research.
- Stability across production conditions
The raw materials used in this study were produced at a GMP-certified facility designated by the Rural Development Administration (RDA), Republic of Korea. The RDA strictly regulates diet composition, developmental stage, and rearing environment to ensure batch-to-batch consistency for P. brevitarsis larvae, as part of their initiative to expand its use as a functional food ingredient. Therefore, there was no variability in these factors across the samples we studied. We analyzed three independent production batches from this facility, and L-tryptophan content showed no significant variation, indicating strong reproducibility under standardized production conditions (Supplementary Materials: Figure S1).
- Sensitivity to degradation
We acknowledge that degradation studies under suboptimal processing or storage conditions were not performed in this work. Because all samples were processed and stored under identical GMP-controlled conditions, there was no opportunity to assess the effect of poor processing on L-tryptophan stability. We agree this is an important area for further study and plan to include degradation experiments in future work.
- Specificity
We recognize that L-tryptophan is a common amino acid and is not unique to P. brevitarsis. However, our repeated fractionation and compositional analyses of GMP-produced P. brevitarsis extracts consistently revealed L-tryptophan as the most abundant low-molecular-weight compound, with minimal variation among batches. Other amino acids or small molecules were present at much lower concentrations and exhibited greater variability, making them less suitable as robust QC markers. We also note that in other functional food materials, such as certain plants, common compounds (e.g., resveratrol, chlorogenic acid) are often used as QC markers when they are abundant, stable, and reproducible. Based on these criteria, we selected L-tryptophan as the QC marker for P. brevitarsis larval extracts.
In conclusion, we sincerely appreciate the reviewer’s thoughtful comments, which have helped us to better recognize the current limitations of our work—particularly the absence of multi-source validation and degradation studies. While the present study focused on establishing a practical quality control marker under Good Manufacturing Practice (GMP) conditions, we fully acknowledge the importance of validating its biological relevance, evaluating its stability under various conditions, and confirming its consistency across diverse sources. We are committed to addressing these aspects in future collaborative studies, and we hope that the revisions made in this manuscript will clarify our rationale for selecting L-tryptophan and convey our willingness to strengthen this line of research. We have revised the manuscript to add these points in Conclusion Section. Please view:
In the present study, L-tryptophan was selected as a quality control (QC) marker for Protaetia brevitarsis seulensis larval extracts produced under Good Manufacturing Practice (GMP) conditions. This selection was based on repeated fractionation and compositional analyses, which consistently revealed L-tryptophan as the most abundant low-molecular-weight compound with minimal batch-to-batch variation across three independent production lots from a GMP-certified facility designated by the Rural Development Administration (RDA), Republic of Korea. While L-tryptophan is a common amino acid and not unique to P. brevitarsis, its high abundance, stability under standardized production conditions, and reproducibility in quantitative analysis make it a practical marker for routine QC testing. In other functional food materials, widely distributed compounds such as resveratrol and chlorogenic acid are similarly used as QC markers when they demonstrate such characteristics. We acknowledge that additional studies are needed to assess its correlation with functional properties, stability under suboptimal processing or storage conditions, and variability across different rearing environments. These aspects will be addressed in future collaborative research.
Small other points:
Title and Abstract
HPLC is not defined in the abstract.
In lines 10 and 21, the full scientific name should be given with the authority and year of description at the first occurrence, along with the family and order, since this insect is not widely known.
Answer: These comments were corrected in the revised manuscript.
Introduction
Line 46: Please clarify that composition can vary depending on species, developmental stage, and nutrition.
Line 50: Mention that chitin can cause allergic reactions, as this is relevant for edible insect products.
Line 55: As above, include the full scientific name with descriptor and year.
Lines 65–67: The sentence is heavy and the English should be revised for clarity.
Line 77: HPLC-DAD is not defined at first mention.
Lines 79–82: This section reads more like results than introduction.
Answer: These comments were corrected in the revised manuscript.
Materials and Methods
Line 112: Explain why the extraction starts from ethanol rather than directly from the powdered sample.
Answer: The raw materials used in this study were produced at a GMP-certified facility designated by the Rural Development Administration (RDA), Republic of Korea. We chose to start the extraction with ethanol rather than directly from the powdered sample because ethanol efficiently extracts bioactive compounds while ensuring safety and suitability for use as a functional food ingredient. Using ethanol also allows for better standardization and reproducibility of the extract, which is essential for both quality control and potential industrial applications.
Line 117: The text implies that this compound is the major one, but there could be 2–3 compounds present in similar proportions. Were preliminary tests performed to justify this assertion? If so, they should be described.
Answer: These comments were corrected in the revised manuscript.
Discussion
The discussion does not address the relevance of L-tryptophan as a marker for quality control. This should be elaborated, with reference to practical or regulatory contexts.
The study did not test samples from different production conditions. Would the tryptophan profile remain stable under varying rearing or processing conditions? This limitation should be acknowledged.
Answer: We have revised the manuscript to add these points in Conclusion Section. Please view:
In the present study, L-tryptophan was selected as a quality control (QC) marker for Protaetia brevitarsis seulensis larval extracts produced under Good Manufacturing Practice (GMP) conditions. This selection was based on repeated fractionation and compositional analyses, which consistently revealed L-tryptophan as the most abundant low-molecular-weight compound with minimal batch-to-batch variation across three independent production lots from a GMP-certified facility designated by the Rural Development Administration (RDA), Republic of Korea. While L-tryptophan is a common amino acid and not unique to P. brevitarsis, its high abundance, stability under standardized production conditions, and reproducibility in quantitative analysis make it a practical marker for routine QC testing. In other functional food materials, widely distributed compounds such as resveratrol and chlorogenic acid are similarly used as QC markers when they demonstrate such characteristics. We acknowledge that additional studies are needed to assess its correlation with functional properties, stability under suboptimal processing or storage conditions, and variability across different rearing environments. These aspects will be addressed in future collaborative research.
Reviewer 2 Report
Comments and Suggestions for Authors
I have gone through the manuscript entitled “Isolation and Quantification of L-Tryptophan from Protaetia brevitarsis seulensis Larvae as a Marker for Quality Control of Edible Insect Extract”. The authors have done a commendable job by selecting Protaetia brevitarsis seulensis larvae, a traditionally used and scientifically promising edible insect. By highlighting L-tryptophan, an essential amino acid crucial for nutrition and brain health, they significantly enhance the study's value. Their thorough analysis reveals the nutritional benefits of this insect, and the establishment of a reliable HPLC method ensures quality and consistency. This research provides a solid groundwork for future investigations and the development of products that utilise insect-based functional ingredients.
After thoroughly reviewing the entire article, I have the following recommendations:
- It would be beneficial for clarity if the authors explicitly report the yield of L-tryptophan as mg per gram of crude extract or as a percentage of the total extract, as mentioned that 70 mg was isolated from 100 g of extract. This would enhance the practical relevance for industrial or commercial applications.
- The manuscript could include a discussion on L-tryptophan's stability, since it can degrade under certain conditions, such as sensitivity to oxidative degradation and specific environmental conditions. Adding information on storage, handling, and stability would make it more reliable for commercial use.
- Although individual HPLC chromatograms are shown, an overlay of the standard, isolated compound, and EtOH and n-BuOH extracts would improve clarity and strengthen identification through retention time comparison.
- The manuscript specifies analytical replicates for HPLC quantification (n = 5), but it doesn't clarify if the extraction and isolation procedures were done on multiple biological replicates or just once. It would be helpful to address this to assess the reproducibility of the extraction process and the consistency of L-tryptophan content across samples.
- The term "Q-Marker" has been mentioned in the abstract, but it lacks a clear definition. Providing a concise definition of Q-marker earlier in the text would be helpful for readers.
Author Response
Reviewers' comments to Author:
I have gone through the manuscript entitled “Isolation and Quantification of L-Tryptophan from Protaetia brevitarsis seulensis Larvae as a Marker for Quality Control of Edible Insect Extract”. The authors have done a commendable job by selecting Protaetia brevitarsis seulensis larvae, a traditionally used and scientifically promising edible insect. By highlighting L-tryptophan, an essential amino acid crucial for nutrition and brain health, they significantly enhance the study's value. Their thorough analysis reveals the nutritional benefits of this insect, and the establishment of a reliable HPLC method ensures quality and consistency. This research provides a solid groundwork for future investigations and the development of products that utilise insect-based functional ingredients.
After thoroughly reviewing the entire article, I have the following recommendations:
It would be beneficial for clarity if the authors explicitly report the yield of L-tryptophan as mg per gram of crude extract or as a percentage of the total extract, as mentioned that 70 mg was isolated from 100 g of extract. This would enhance the practical relevance for industrial or commercial applications.
Answer: These comments were corrected in the revised manuscript.
The manuscript could include a discussion on L-tryptophan's stability, since it can degrade under certain conditions, such as sensitivity to oxidative degradation and specific environmental conditions. Adding information on storage, handling, and stability would make it more reliable for commercial use.
Answer: Thank you for your valuable comment. The raw materials used in this study were produced in a GMP-certified facility, and we are currently conducting additional experiments within the GMP facility to assess their stability. We will include the results and discussion on L-tryptophan stability, storage, and handling in future studies
Although individual HPLC chromatograms are shown, an overlay of the standard, isolated compound, and EtOH and n-BuOH extracts would improve clarity and strengthen identification through retention time comparison.
Answer: Overlay spectrum was added in Supplementary Materials (Figure S3).
The manuscript specifies analytical replicates for HPLC quantification (n = 5), but it doesn't clarify if the extraction and isolation procedures were done on multiple biological replicates or just once. It would be helpful to address this to assess the reproducibility of the extraction process and the consistency of L-tryptophan content across samples. The raw materials used in this study were produced at a GMP-certified facility designated by the Rural Development Answer: Administration (RDA), Republic of Korea. The RDA strictly regulates diet composition, developmental stage, and rearing environment to ensure batch-to-batch consistency for P. brevitarsis larvae, as part of their initiative to expand its use as a functional food ingredient. Therefore, there was no variability in these factors across the samples we studied. We analyzed three independent production batches from this facility, and L-tryptophan content showed no significant variation, indicating strong reproducibility under standardized production conditions (Supplementary Materials: Figure S1).
The term "Q-Marker" has been mentioned in the abstract, but it lacks a clear definition. Providing a concise definition of Q-marker earlier in the text would be helpful for readers.
Answer: Thank you for your comment. We agree that a clear definition of the term ‘Q-marker’ is important. We have now added a concise definition earlier in the text to help readers understand its meaning and relevance in the context of quality control.
Reviewer 3 Report
Comments and Suggestions for Authors
The communication entitled "Isolation and Quantification of L-Tryptophan from Protaetia brevitarsis seulensis Larvae as a Marker for Quality Control of Edible Insect Extract" is well-structured and presents valuable findings regarding the analysis of L-tryptophan, a component of the Protaetia brevitarsis seulensis larvae extract, proposed as a potential quality marker. The overall structure is logical, the results are solid, and the methods used are appropriate. Only a few aspects could be improved. My comments are provided below.
Abstract:
Lines 21–22: Please rephrase the sentence as it contains two uses of "have" and multiple repetitions of the word "traditional."
Lines 24–26: Use a more formal and objective tone, avoid first-person plural ("we").
Methods:
Based on the results, the authors mention NMR analysis, but this technique is not described in the Methods section. A brief description of the NMR procedure would be beneficial.
Lines 101–107: This section should be separated into a new paragraph with an appropriate heading, such as "Chemicals."
Lines 118–121: This part is unclear and requires clarification. Could you please specify the volume used, the number of replicates, and the exact mass being referred to?
Lines 131–137: The explanation here should be improved and expanded, as this part is better described later in the Results section.
Figures and Tables:
Scheme 1: Please add a list of abbreviations below the scheme to help the reader understand the symbols and terms used.
Table 1: Please correct "LDQ" to "LOQ."
Please improve the resolution and visual quality of Figures S3 and S4.
Author Response
Reviewers' comments to Author:
The communication entitled "Isolation and Quantification of L-Tryptophan from Protaetia brevitarsis seulensis Larvae as a Marker for Quality Control of Edible Insect Extract" is well-structured and presents valuable findings regarding the analysis of L-tryptophan, a component of the Protaetia brevitarsis seulensis larvae extract, proposed as a potential quality marker. The overall structure is logical, the results are solid, and the methods used are appropriate. Only a few aspects could be improved. My comments are provided below.
Abstract:
Lines 21–22: Please rephrase the sentence as it contains two uses of "have" and multiple repetitions of the word "traditional."
Answer: These comments were corrected in the revised manuscript.
Lines 24–26: Use a more formal and objective tone, avoid first-person plural ("we").
Answer: These comments were corrected in the revised manuscript.
Methods:
Based on the results, the authors mention NMR analysis, but this technique is not described in the Methods section. A brief description of the NMR procedure would be beneficial.
Answer: Thank you for your valuable comment. A description of the NMR analysis and the associated methodology has been added to the Methods section.
Lines 101–107: This section should be separated into a new paragraph with an appropriate heading, such as "Chemicals."
Answer: These comments were corrected in the revised manuscript.
Lines 118–121: This part is unclear and requires clarification. Could you please specify the volume used, the number of replicates, and the exact mass being referred to?
Answer: These comments were corrected in the revised manuscript.
Lines 131–137: The explanation here should be improved and expanded, as this part is better described later in the Results section.
Answer: These data correspond to the NMR analysis of the isolated compound. The complete NMR data are provided to allow readers to review the structural information conveniently. The discussion later in the Results section presents the interpretation of the provided NMR data.
Figures and Tables:
Scheme 1: Please add a list of abbreviations below the scheme to help the reader understand the symbols and terms used.
Answer: These comments were corrected in the revised manuscript.
Table 1: Please correct "LDQ" to "LOQ."
Answer: These comments were corrected in the revised manuscript.
Please improve the resolution and visual quality of Figures S3 and S4.
Answer: These comments were corrected in the revised manuscript (Supplementary Materials: Figure S5 and S6).
Round 2
Reviewer 1 Report
Comments and Suggestions for Authors
Thank you for adressing my comments and add limitations in the conclusion section.